# Further Characterization of Rio Grande Virus and Potential for Cross Reactivity with Rift Valley Fever Virus Assays

**DOI:** 10.3390/v13091719

**Published:** 2021-08-30

**Authors:** Mitchell S. Szymczak, Will K. Reeves, Myrna M. Miller

**Affiliations:** 1College of Osteopathic Medicine of the Pacific, Western University of Health Sciences, Pomona, CA 91766, USA; mitchell.szymczak@westernu.edu; 2C.P. Gillette Museum of Arthropod Diversity, Colorado State University, Fort Collins, CO 80523, USA; wkreeves@gmail.com; 3Wyoming State Veterinary Laboratory, University of Wyoming, Laramie, WY 82070, USA

**Keywords:** Rio Grande virus, phlebovirus, sand-fly-transmitted, southern plains woodrat, antigenic cross-reaction, Rift Valley fever virus, United States

## Abstract

Phleboviruses (genus *P**hlebovirus*, family *Phenuiviridae*) are emerging pathogens of humans and animals. Sand-fly-transmitted phleboviruses are found in Europe, Africa, the Middle East, and the Americas, and are responsible for febrile illness and nervous system infections in humans. Rio Grande virus (RGV) is the only reported phlebovirus in the United States. Isolated in Texas from southern plains woodrats, RGV is not known to be pathogenic to humans or domestic animals, but serologic evidence suggests that sheep (*Ovis aries*) and horses (*Equus caballus*) in this region have been infected. Rift Valley fever virus (RVFV), a phlebovirus of Africa, is an important pathogen of wild and domestic ruminants, and can also infect humans with the potential to cause severe disease. The introduction of RVFV into North America could greatly impact U.S. livestock and human health, and the development of vaccines and countermeasures is a focus of both the CDC and USDA. We investigated the potential for serologic reagents used in RVFV diagnostic assays to also detect cells infected with RGV. Western blots and immunocytochemistry assays were used to compare the antibody detection of RGV, RVFV, and two other New World phlebovirus, Punta Toro virus (South and Central America) and Anhanga virus (Brazil). Antigenic cross-reactions were found using published RVFV diagnostic reagents. These findings will help to inform test interpretation to avoid false positive RVFV diagnoses that could lead to public health concerns and economically costly agriculture regulatory responses, including quarantine and trade restrictions.

## 1. Introduction

Viruses within the genus *Phlebovirus*, family *Phenuiviridae*, include mosquito-, sand fly-, and tick-borne pathogens, and these viruses have three negative sense RNA segments (4). The first phlebovirus identified in the United States, Rio Grande virus (RGV), was isolated from southern plains woodrats (*Neotoma micropus*) in Texas [1,2], and was shown to be transmitted by a sand fly, *Lutzomyia anthophora,* in the laboratory [3]. Some domestic animals, including horses (*Equus cabalus*) and sheep (*Ovis aries*), were found to be antibody-positive, indicating prior infection. No antibodies were found in 278 human serum samples tested, and sera from native wild ruminants were not tested [1]. Rio Grande virus is phylogenetically related to Anhanga virus, and is also grouped with Tapara virus in analyses based on the nucleocapsid protein (NP) [4]. *Neotoma micropus*, thought to be the natural host, ranges across North America from southeastern Colorado, east to central Kansas, west to the Arizona border, and south to the eastern coast of Mexico in northern Veracruz [5]. The probable insect vector is *L. anthophora*, a sand fly found in or near rodent and lagomorph nests in southern Texas and northern Mexico [6].

Rift Valley fever virus (RVFV), an African mosquito-borne phlebovirus, causes new born fatalities and abortions in ruminants, and potentially fatal hemorrhagic disease in humans. This zoonotic virus can be transmitted to humans by direct contact with meat or fluids from infected animals, or from an infected mosquito, and has been identified as a top concern by the CDC and the USDA [7]. Diagnostic testing and vaccine development for RVFV are currently research priorities in the United States and around the world. Recently, white-tailed deer (*Odocoileus virginianus*) were found to be susceptible to RVFV infection, potentially serving as a reservoir and amplification host [8]. Due to the abundance and geographic distribution of these animals, they have been proposed as sentinels for RVFV and are likely to play a major epidemiologic role if the virus were introduced into the United States.

Identification of newly discovered viruses was previously based on virus serum neutralization assays which take advantage of cross-reacting antibodies between closely related viruses [9]. It is important for diagnostic test interpretation to identify and be aware of the potential for false positive interpretations of the assay. The potential for a Toscana virus nucleocapsid protein diagnostic assay to detect divergently related viruses within the genus *Phlebovirus* was previously reported [10]. The assay produced positive results for Aguacate virus, Anhanga virus, Chagres virus, and Punta Toro virus (South and Central America), and Arumowot virus (Africa), but not RGV or RVFV. Antigenic cross-reactions between related viruses can result in misdiagnosis if this potential is not recognized. This was illustrated by the 1999 introduction of an exotic arbovirus, West Nile virus, into New York, USA. The initial human cases were misdiagnosed as St. Louis encephalitis, a flavivirus enzootic to North America, based on positive enzyme-linked immunosorbent assays (ELISA) [11].

Misinterpretation of false positive results in diagnostics has the potential for significant consequences. Falsely identifying a foreign animal disease, such as RVFV, is likely to lead to regulatory responses including farm and ranch quarantines, trade restrictions, and elevated public health concern. Our research looked for potential for cross-reactions between RGV and RVFV based on diagnostic reagents used to detect virus antigens or antibodies.

## 2. Materials and Methods

**Viruses**. RGV isolate TBM3-204 was provided by Dr. R. McClain, USDA-APHIS Wildlife Service, National Wildlife Research Center. The attenuated RVFV strain, MP-12, was obtained from the U.S. Army Medical Research Institute of Infectious Diseases (USAMRIID), as previously published [12]. Punta Toro virus and Anhanga virus were acquired from ATCC (Manassas, VA, USA). Viruses were propagated in Vero 76 (V76) cells (ATCC), grown with complete media consisting of modified Eagle’s medium (Corning, Manassas, VA, USA), 100 U/mL penicillin and 100 ug/mL streptomycin sulfate (Lonza, Walkersville, MD, USA), and supplemented with 4% fetal bovine serum (VWR, Radner, PA, USA). The titers of the virus stocks were 10^6.41^, 10^6.50^, 10^6.29^, and 10^7.14^ tissue culture infectious dosage 50% (TCID50) for Anhanga, Punta Toro, RGV, and MP-12, respectively.

**Antibodies**. Primary anti-RVFV antibodies used in this study included sera from sheep vaccinated with MP-12 [12], and rabbits vaccinated with RVFV NP protein [13]. Secondary HRP-conjugated antibodies used in Western blots were donkey anti-sheep IgG-HRP (R&D Systems, Minneapolis, MN, USA), or mouse anti-rabbit IgG-HRP (Santa Cruz Biotechnology, Dallas, TX, USA). Fluorescent labeled secondary antibodies for immunocytochemistry were donkey anti-sheep IgG-NL493 or donkey anti-rabbit IgG-NL493 or (R&D Systems, Minneapolis, MN, USA).

**Immunocytochemistry.** Indirect immunofluorescence testing was applied to noninfected V76 cells, or cells infected with Anhanga virus, Punta Toro virus, RGV, or MP-12 grown on microwell slides at a multiplicity of infection of 0.5. Cells were fixed with acetone when cytopathic effects were first observed, which ranged from 2 days post-infection for MP-12, to 4–5 days post-infection for Anhanga virus, Punta Toro virus, and RGV. Fixed cells were incubated with rabbit anti-RVFV NP protein or sera from MP-12-vaccinated sheep, diluted 1:50 in PBS. Secondary fluorescent labeled antibodies, diluted 1:1000 in PBS, were used to detect infected cells. Nuclei were stained with DAPI (Dapi Fluoromount-G, SouthernBiotech, Birmingham, AL, USA) and actin filaments were detected with rhodamine-conjugated phalloidin (Thermo Fisher Scientific, Waltham, MA, USA). Images were captured using an EVOSfl fluorescence microscope (Life Technologies Corp., Carlsbad, CA, USA).

**Protein detection by Western blotting.** Standard SDS-PAGE methods were used for protein detection [14,15]. Briefly, proteins were extracted from V76 cells infected with MP-12, RGV, Anhanga virus, or Punta Toro virus when exhibiting 50–70% cytopathic effect, which generally occurred by day 3 post-infection for MP-12 and day 5–7 for RGV, Anhanga virus and Punta Toro virus. Protein from noninfected V76 cells served as the negative control. Protein bands were probed with rabbit anti-RVFV NP or sera from sheep vaccinated with MP-12, diluted 1:500 in PBS with 0.1% Tween, followed by secondary antibodies diluted 1:1000 in PBS, and detected with TMB-stabilized substrate for HRPO (Promega, Madison, WI, USA).

**Whole-Genome Sequencing**. Whole-genome sequencing (WGS) was performed to verify the identity of the viruses used in this study, and to obtain the WGS of RGV. Briefly, the viruses were grown in 75 cm flasks and frozen at −80 °C when >80% CPE was observed. Cultures were thawed and clarified by centrifugation at 5000× *g* for 30 min, and virus particles were semi-purified by centrifugation through a 20% sucrose cushion at 105,000× *g* for 12 h. The pellet was resuspended in 400 μL of PBS and RNA extracted with Trizol reagent (Life Technologies), following the manufacturer’s recommendation. Sequencing was performed at the University of Illinois Keck Center using the Illumina MiSeq (Illmina, Inc., San Diego, CA, USA) platform. Sequencing reads were assembled using Ray de novo assembly [16], and contigs greater than 1500 nt were searched using NCBI Blast.

**Measurement of Fluorescence Intensity**. The fluorescent intensity of granules from MP-12- and RGV-infected cells was measured using ImageJ software [17]. Four areas from each image were measured, and the means and standard deviations were calculated. Differences in intensity between viruses were tested using the Student’s *t*-test, with significance set at *p* < 0.05.

## 3. Results

### 3.1. Immunocytochemistry

Immunofluorescent assays using anti-RVFV antibody reagents were used to determine their potential to also detect RGV-infected cells. Cells infected with the RVFV-attenuated strain, MP-12, stained with rabbit anti-NP primary antibody, exhibited strong fluorescence staining of abundant cytoplasmic granules, as seen in Figure 1a. Fluorescent cytoplasmic granules were also observed in RGV-infected cells, but were less abundant and detected in fewer cells (Figure 1b). The mean intensities of fluorescent granules from MP-12- and RGV-infected cells (144, SD 7.7, and 127, SD 21.7) were not significantly different. Fluorescent staining was not detected in control cells (Figure 1c), or cells infected with Punta Toro virus or Anhanga virus.

### 3.2. Protein Detection Using Western Blots

Western blots were used to identify proteins from virus-infected cells. When probed with rabbit anti-NP, bands of the expected size of 27.4–27.8 kD were identified in protein extracts from cells infected with MP-12, RGV, and Anhanga virus (Figure 2). Western blots performed with sera from sheep vaccinated with MP-12 had more background staining and distinct bands were not detected

### 3.3. Whole-Genome Sequencing

Whole-genome sequencing was performed to verify the identity of viruses used in this study, and to obtain WGS for RGV. Complete segments were obtained for RGV and were placed in GenBank with accession numbers MK503253 (L), MK503254 (M), and MK503255 (S). The identity of MP-12, Anhanga virus, and Punta Toro virus were verified by NCBI nucleotide Blast of complete sequences of all segments, with 99.2–100% identity.

## 4. Discussion and Conclusions

Rio Grande virus is the only known phlebovirus endemic to the United States, and the potential for cross-reactivity with RVFV has not previously been reported. The USDA has identified RVFV as the most significant arthropod-borne animal disease threat to U.S. livestock. Due to this concern, the development of diagnostic assays to detect RVFV or anti-RVFV antibodies has been a priority of the USDA, and APHIS National Veterinary Stockpile committee. The risk of RVFV introduction to the Western hemisphere is not limited to the United States. Numerous native phleboviruses are endemic to Central and South America [18,19], and serologic cross-reactions could complicate diagnostics of RVFV in these areas.

We tested the ability of antibodies developed to detect RVFV proteins to also detect proteins from RGV, Anhanga virus, and Punto Toro virus. Cross-reactions with RGV were identified by immunocytochemistry and Western blotting, which also detected Anhanga virus. This is a preliminary study, and further research is needed to verify and determine the extent of antigenic cross-reactions. Limitations of the study include the need to test using methods commonly used for high-throughput diagnostics, such as ELISA and fluorescence microsphere immunoassays. Such assays have been developed for diagnoses of RVFV [20,21,22,23,24], but none are currently commercially available in the United States. Examination and testing of NP amino acid coding sequence for regions of shared antigenic identity would be helpful in developing specific assays. There is a need for challenge infections of relevant ruminant species, including white-tailed deer, to determine host susceptibility and produce valuable anti-RGV antibodies for laboratory studies. Additionally, on the population level, assessing the prevalence and distribution of natural RGV infections in domestic and wild ruminant species is needed to determine the real potential for false positive reactions.

The results of this study suggest that some diagnostic tests designed to detect RVFV proteins in U.S. animals could give false positive results in RGV-endemic regions. Rio Grande virus is currently only known to occur in Texas, but the range of the southern plains woodrat includes Colorado, Kansas, New Mexico, and much of Mexico, and the sand fly vector species ranges from Texas to Alberta, Canada [25]. The likely distribution of RGV closely matches predicted high-risk areas if RVFV were to be introduced into North America [26]. Sheep and horses have been identified as having been infected with RGV [1], and interpretation of positive RVFV test results in these species should be considered potential RGV cross-reactions. Other phleboviruses present in Central and South America may present similar challenges. These findings point to the need for awareness of the potential for false positive test interpretations, and specificity testing of commercial assays against related native viruses.

## Figures and Tables

**Figure 1 viruses-13-01719-f001:**
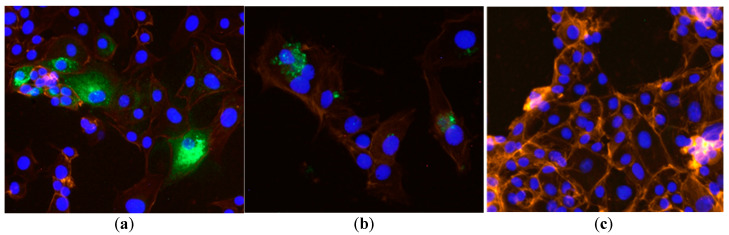
Indirect immunofluorescent staining of V76 cells was used to identify virus-infected cells. Cells infected with the MP-12 vaccine strain of Rift Valley fever virus (**a**), Rio Grande virus (**b**), or uninfected control cells (**c**) were grown on microwell slides prior to fixing with acetone. Fixed cells were stained with rabbit anti-RVFV nucleocapsid protein that was detected with green fluorescent labeled secondary antibody. Nuclei were detected with DAPI (blue), and actin filaments were detected with rhodamine-conjugated phalloidin. Intensely green-stained cytoplasmic granules were observed in MP-12- and RGV-infected cells, but absent from control cells. Images were captured at 40× magnification.

**Figure 2 viruses-13-01719-f002:**
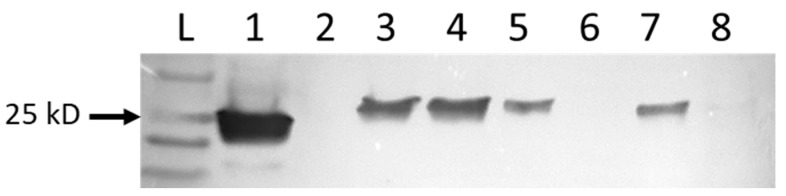
Western blot of denatured proteins extracted from noninfected control V76 cells (lane 6), or infected with the MP-12 vaccine strain of Rift Valley fever virus (lane 1), RGV (lanes 3–5), Anhanga virus (lane 7), or Punta Toro virus (lane 8). Lane 2 was left open to ensure no spillover from the positive MP-12-infected cells. Protein bands were identified with the primary antibody rabbit anti-RVFV nucleocapsid protein, and the secondary antibody mouse anti-rabbit-IgG-HRP. The arrow indicates the 25 kD band of the protein ladder. Proteins of the expected size were identified for MP-12, RGV, and Anhanga virus, but were absent from control cells and those infected with Punta Toro virus.

## Data Availability

Not applicable.

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
