# Peer review of "Further Characterization of Rio Grande Virus and Potential for Cross Reactivity with Rift Valley Fever Virus Assays"

_viruses, 2021, doi:10.3390/v13091719_

Round 1
Reviewer 1 Report
The authors investigate serological reagents used for RVFV diagnosis to cross-react with other closely related phleboviruses. Specifically, MP12 immunized sheep and RVFV-NP immunized rabbit sera was used for detection by IFA and WB with cells infected (or cell lysates) with RVFV MP12, RGV, Anhanga and Punta Toro viruses. Cross-reactivity was shown with RGV by both methods.
Topic is relevant, important, and of particular interest given the impact of a false positive test for RVFV.
Would have liked to see if RGV antibodies or immune sera reacts with RVFV infected cells or antigens.
Minor comments:
Line 45: major
Line55: flavivirus enzootic
Line 58: RVFV (already defined previously)
Line 134: RGV (already defined previously)
Author Response
Thank you for your helpful comments. Please see attachment for our point -by-point response.
Myrna

Reviewer 2 Report
The study explored an intriguing issue related to preparation for a potential emerging of RVFV, a significant pathogen. The method, results and discussion are clear and adequate. The work is very preliminary, therefore having more discussion on further study and interpretation of the data will be useful.
Line 77 and 84, specify the viral titer used in the experiment
Line 78 and 85, specify the days after infection when the cells were prepared for the detections.
Results: Compare NP amino acid sequences for the viruses tested in the study to explore the possible motifs that may correlated to the cross reactivities.
Discussion: it is important to discuss the limitation of the study, possibility of flaws and artifacts, and suggest further work to consolidate the observation.
Reviewer 3 Report
The topic of this paper is timely and the hypothesis needs to be tested. The New World is not prepared for acurate diagnosis of Rift Valley Fever since our endemic phleboviruses have not been evaluated for cross-reactivity with Rift. The authors made a good start to this end but improvements are needed to definitively say if there is any significant cross-reactivity between Rift and new world pleboviruses.
Cross reactivity is always seen in closely related viruses and most available assays (commercial and military) have accounted for this.
- The introduction should include text regarding cross-reactivity of pleboviruses with commercially available Rift assays. It is likely in the event of a new world outbreak that these assays would be used.
- The diagnostic reagents (line 60) were only used against 2 strains of rift and would likely not be used in the event of a new world outbreak since USAMRIID and the EU have reagents in house.
- Methods: define "V76"
- Methods: an ELISA needs to be included for your cross-reactivity analysis to quantify how much general cross-reactivity there is. If possible, use a commercial kit.
- Methods: a plaque reduction neutralization test should be performed as a secondary method to help quantify if there is cross-reactivity since PRNTs are used to confirm exposure.
- Do diagnostic PCR primers for Rift cross react with new world phleboviruses?
- Results: please quantify your fluorescence of your images and run statistics to determine if there is legitimate cross-reactivity. Olympus CellSens does this.
